# Sizing the Knowledge Gap in Taxonomy: The Last Dozen Years of Aphidiinae Research

**DOI:** 10.3390/insects13020170

**Published:** 2022-02-05

**Authors:** Andjeljko Petrović

**Affiliations:** Institute of Zoology, Faculty of Biology, University of Belgrade, Studentski trg 16, 11000 Belgrade, Serbia; andjeljko@bio.bg.ac.rs

**Keywords:** taxonomy, taxonomic impediment, Aphidiinae, parasitoids

## Abstract

**Simple Summary:**

Taxonomy is a biological discipline with the task to identify, name, and describe organisms, and as such, it provides necessary data for all other biological disciplines. The biodiversity crisis through which we are living draws attention to the crucial role of taxonomy in biology today. At the same time, the scientific community, as well as society in general, has become more aware of the difficulties associated with taxonomy, such as gaps in taxonomic knowledge, a lack of taxonomic infrastructure, and an insufficient number of taxonomic experts (“taxonomic impediment”). With this study, we tried to size this knowledge gap by analyzing the taxonomical studies on Aphidiinae (Hymenoptera: Braconidae) conducted from 2010 to 2021. Aphidiinae are endoparasitoids of aphids; a single specimen completes its development inside the living aphid host, which are used in biological control programs. Here, we summarize the knowledge gathered over the last dozen years and discuss it in a general context.

**Abstract:**

Taxonomic impediment is one of the main roadblocks to managing the current biodiversity crisis. Insect taxonomy is the biggest contributor to the taxonomic impediment, both in terms of the knowledge gap and the lack of experts. With this study, we tried to size the knowledge gap by analyzing taxonomical studies on the subfamily Aphidiinae (Hymenoptera: Braconidae) conducted from 2010 to 2021. All available taxonomic knowledge gathered in this period is critically summarized: newly described species, detection of alien species, published identification keys, etc. All findings are discussed relative to the current state of general taxonomy. Future prospects for taxonomy are also discussed.

## 1. Prologue

The biodiversity crisis has been known about for decades, but has just recently started drawing the public attention it deserves [1]. There is an urgent need to mitigate the crisis by exploring, managing, and conserving biodiversity. The first step is to realize that the most important unfinished job in biology is discovering and describing biodiversity [2,3]. This brings us to the vital role of taxonomy in today’s biology. As a biological discipline with the tasks of identifying, naming, and describing organisms [4], taxonomy represents a bridge between two basic biological disciplines, morphology and systematics, and provides necessary data for all other biological disciplines (Figure 1). The strongest link between taxonomy and systematics/morphology is in phylogenetic taxonomy, which uses data about common ancestry from systematic (and phylogenetic) studies (based mainly on morphology and/or molecules) and combines it with morphological data in species descriptions. To the untrained eye, taxonomy might look like an easy job, but it must collate information from various scientific fields (e.g., morphology, anatomy, ecology, molecules, geography, etc.), which makes it the most integrative biological discipline [5,6] (Figure 1). Unfortunately, at the moment, taxonomy is not the highway bridge that we need. It is like an old, decrepit bridge that is still standing on a good foundation, but is full of gaps, holes, and obstacles. The difficulties associated with taxonomy are gaps in taxonomic knowledge, a lack of taxonomic infrastructure, and an insufficient number of experts; together these are called the “taxonomic impediment”. With more than 1 million described species, insects comprise more than half of all known species [7], and with an estimate of 5.5 million living species [8], they represent the biggest proportion of the taxonomic impediment. 

Knowledge gaps in taxonomy are present for all insect groups, differing only in size. Generally, the size of the gap is negatively correlated with body size and positively correlated with the number of species in the group. Families of small Diptera and parasitoid wasps (Hymenoptera) are recognized as groups with the highest proportions of dark taxa (little-known and unknown species) [9,10,11].

In order to present the size of the knowledge gap in taxonomy, the subfamily Aphidiinae (Hymenoptera: Braconidae) is chosen as an example. 

All species belonging to the subfamily Aphidiinae are solitary koinobiont endoparasitoids of aphids and, as such, a single specimen completes its development inside the living aphid host, which continues to feed and grow [12]. Aphidiinae are highly specialized to attack only aphids and several species are widely used as biocontrol agents. Primarily because of their economic importance, Aphidiinae are one of the best studied parasitoid groups. The taxonomy and systematics of Aphidiinae are especially well studied in Europe, with over 70 years of continuous research [13]. The history of Aphidiinae taxonomy dates back to the dawn of nomenclature and taxonomy, with the first species described in the 10th edition of *Systema Naturae* (*Aphidius rosae* Haliday, 1834 described as *Ichneumon aphidum* L. 1758) [14]. In the last 263 years, many notable entomologists have contributed to Aphidiinae taxonomy (e.g., Christian Gottfried Daniel Nees von Esenbeck, Alexander Henry Haliday, Thomas Ansell Marshall, William Harris Ashmead, and many others), but the real pioneers are our contemporaries: Professor Manfred Mackauer (Simon Fraser University, Burnaby, Canada) and Dr. Petr Starý (Biology Centre of the Czech Academy of Sciences, České Budějovice, Czech Republic). More than six decades ago, they started their research on Aphidiinae, and have published numerous papers concerning taxonomy, systematics, and all other aspects of Aphidiinae biology. Dr. Petr Starý, with more than 500 published papers dealing with Aphidiinae, can undoubtedly be labeled as a man who made a difference. One may ask why there is a knowledge gap when one man has performed so much research. The answer would be that taxonomy is a never-ending story; so, here we present taxonomical studies on Aphidiinae conducted from 2010 to 2021, as a proxy of the knowledge gap size.

## 2. What Has Been Accomplished in the Last Dozen Years?

The first problem that emerges when one starts reading about the subfamily Aphidiinae is the number of described species and genera. Those numbers differ significantly in various sources, starting with 400 species in 50 genera in Boivin et al. [12], then 505 species in 38 genera according to Žikić et al. [15], up to more than 600 species in 65 genera according to Tian et al. [16], and 700 species according to Mackauer and Finlayson [17]. There are several reasons for this discrepancy, such as the use of outdated references (the last comprehensive world checklists of Aphidiinae were published in the late 1960s [13,18]); uncritical use of available databases such as Taxapad World Ichneumonidae database [19]; counting both recent and fossil species, etc. The determination of the exact number of Aphidiinae species is beyond the subject of this paper, but according to the available data (critical use of data from Taxapad World Ichneumonidae [19] and Fauna Europea [20], combined with Starý 2006 [21] and references therein, and references from this study), our best estimate is that there were about 500 living species classified in 52 genera prior to 2010.

### 2.1. Bookworm on the World Wide Web

The subfamily Aphidiinae is an excellent model to emphasize the “taxonomic impediment.” It is a group of insects that are frequently used for different studies. The aim of the literature survey was to identify as many studies as possible on Aphidiinae, including articles, books, book chapters, conference proceedings, Master’s and PhD theses, research reports, etc., and to identify all new Aphidiinae taxa described in 2010–2021. Literature surveys face specific problems in these times of a constantly growing volume of research. Choosing the right tool for the survey is essential, and there are numerous studies that assess the usefulness of different search systems ([22,23,24] and references therein). In most studies, search engines (such as Google Scholar, Microsoft Academia, etc.) are not recommended to be used solely for literature surveys [22,23], but there are some studies that showed that Google Scholar (GS) can at least be used (with some limitations and extra labor) as a data source for research assessment [24].

In order to determine how many studies dealing with Aphidiinae were published in the last 12 years (2010–2021) a bibliographic search was performed using Web of Science Core Collection (WoS), Scopus, and GS. In all three search systems, the following words and word combinations were used as descriptors for the search: Aphidiinae, Aphidiidae, and aphid parasitoids in 2010–2021. All the descriptors were searched throughout all fields (the whole article without references). Considering that the relevance-based sorting algorithm of GS provides only 1000 results per query (biased towards highly cited documents) [22,24], independent searches were performed for every pair of consecutive years (2010–2011, 2012–2013, … 2020–2021). In this way, all search queries had fewer than 1000 results and bias was eliminated. In all three search systems, the following methodology was used for article selection: 

(1) All search results were inspected by eye and those studies with at least one of the descriptors appearing in the title, keywords, or abstract were treated as relevant. (2) All studies that did not meet the first criterion were further inspected by checking the Materials and Methods and Results sections of research articles, and the full text of all other types of studies. Studies in which Aphidiinae were identified as an object of the study were treated as relevant. (3) Studies that did not meet the previous two criteria were excluded. (4) Studies written in languages unfamiliar to the author were automatically translated with Google Translate (although the translation is not always perfectly accurate, it can be easily used for all detected languages if the one who is using it is familiar with the subject). (5) Obtained sets of studies were then checked and all duplicates were excluded (only for GS results; Scopus and WoS searches resulted in no duplicate results). (6) Results that matched the previous selection criteria were exported as .csv (excel) files. In order to export results from GS, every individual result was manually added to a library and then exported. (7) Obtained datasets had different structures, and it was not possible to compare them automatically (mainly because of the unusual structure of the .csv file exported from GS), so it was completed manually. Every study title obtained in Scopus and WoS searches (which both resulted in a significantly smaller number of results) was searched for again in GS, and if it had already been added to the library, it was treated as a duplicate; those that were not in the library were treated as unique results and were added to the final dataset.

Search results differed significantly between WoS, Scopus, and GS (Table 1), which is as expected based on the differences in their coverage. GS, as the most comprehensive academic search engine (with over 300 million records) [25], provided the highest number of both initial and relevant results. 

After a final comparison of datasets obtained from different search systems, we had a total set of 1902 studies published on Aphidiinae in 2010–2021. Interestingly, in the GS search, 1752 records were acquired, so only 150 of the total records were omitted. An additional GS search (a search of individual articles) resulted in finding all 150 previously unrecorded articles (from Scopus and WoS). Although it remains unknown why those results were omitted in the initial search, it can be concluded that GS has 100% coverage of Aphidiinae studies from the analyzed time period. 

Among the 1902 analyzed results, the biggest proportion (86%) were papers published in scientific journals, but there was also a significant number (14%) of conference papers, books, book chapters, Master’s and PhD theses, etc. (almost all recorded just by GS search). Besides the difference in non-journal-article records, the GS, Scopus, and WoS search results differed significantly in terms of the number of records published in languages other than English. There is a small number of non-English-language journals indexed in both Scopus and WoS (all with abstracts in English), and thus a GS search recovered 292 results in 19 languages other than English, while Scopus detected 22 results in six languages, and WoS detected only 17 in five languages. 

Based on the Aphidiinae literature survey results obtained with GS, Scopus, and WoS, it is obvious that the sole use of curated databases, such as WoS and Scopus, is inadequate for reviews of taxonomical and faunistic research. Although GS receives (mainly deserved) criticism for not being suitable as a primary tool for systematic reviews (because of inadequate recall, precision, transparency, and reproducibility) [22], in this particular case (a survey of Aphidiinae studies), it has been shown that a specifically performed search (as described above) makes GS suitable for a survey of taxonomy literature (with 100% coverage in this study). This search was performed only for the subfamily Aphidiinae, but it will most likely provide similar results for the majority of insect groups because of the specific nature of taxonomic publications. Insect taxonomists often publish their studies in journals that are not indexed in databases. Some of the journals are local, while some are well-respected journals with a long tradition. For example, Труды Русскoгo энтoмoлoгическoгo oбщества (Proceedings of the Russian Entomological Society, published since 1861), The Entomologist’s Monthly Magazine (published since 1864), Entomologisk tidskrift (Entomological Journal, published since 1880), Entomofauna (published since 1980), Insecta Mundi (published since 1985), and many more are not indexed in either Scopus or WoS. This “old-fashioned” method of literature searching (checking every article by eye, which resembles a search in the library) might appear time-consuming and labor-intensive, but provides the most comprehensive data. 

### 2.2. New Taxa in the Old World and All Other Worlds

Within the analyzed search results, a vast majority of studies have been conducted on various applied aspects of Aphidiinae biology (life history, demography, functional response, host preference, foraging behavior) and, to a lesser extent, on local fauna, all with the aim of using those parasitoids in biological control. There were fewer than 60 papers focusing on Aphidiinae taxonomy and systematics, and only 42 in which new Aphidiinae taxa are described [16,17,26,27,28,29,30,31,32,33,34,35,36,37,38,39,40,41,42,43,44,45,46,47,48,49,50,51,52,53,54,55,56,57,58,59,60,61,62,63,64,65] (Appendix A and Figure 2). The obvious discrepancy between applicative and taxonomical studies can be treated as a proxy of the “taxonomic impediment” in Aphidiinae taxonomy. Within those 42 papers, four new genera (*Choreopraon* Mackauer, 2012; *Sergeyoxys* Davidian, 2016; *Astigmapraon* Tian et Chen, 2017, *Ishtarella* Martens, 2021) [16,17,43,65] and 64 new species were described around the world (Appendix A). 

Newly described taxa represent 7% and 11% of all known Aphidiinae genera and species, respectively (Figure 2). 

The fact that such a high percentage of all known Aphidiinae species have been described just in the last 12 years indicates that there are potentially many more species waiting to be found, and thus the knowledge gap in Aphidiinae taxonomy is still large. The assumption can be rightly criticized that the only way to determine the knowledge gap is to estimate the total number of species, and determine the ratio of described to undescribed. Estimating the species number of any insect group, as well as of insects in general, is challenging for many reasons. There are numerous datasets and methodologies that can be used [8]. There are also numerous assumptions that must be made and thus, the obtained results can vary significantly. So, let us try to justify the previous assumption with the simplest and probably most conservative estimation of the number of Aphidiinae species. For this purpose, the parasitoid–host (P:H) ratio can be used, because Aphidiinae are obligatory aphid parasitoids. The main idea is to calculate P:H for some representative areas or territories that are characterized by well-researched aphid and Aphidiinae fauna, and then calculate the number of Aphidiinae species based on the total number of aphid species. Based on the fact that the majority of aphid species are known from the Palaearctic region [66], it is logical to look for representative areas in this realm. The only two areas with up-to-date and relatively well researched faunas of both aphids and parasitoids are the Czech Republic and Serbia. The Czech Republic is a Central European country with 755 recorded aphid species and 135 (between 130–140) Aphidiinae species [21]. Serbia is a South European country with 385 recorded aphid species [67,68] and 121 Aphidiinae species [69]. The calculated parasitoid–host ratio for the Czech Republic is P:H = 0.18 and for Serbia it is P:H = 0.31. Taking the mean P:H ratio (0.245) and multiplying it by the number of currently known aphid species, which is around 5000 [70], gives us a rough, conservative estimate of 1225 Aphidiinae species. Taking into account that the assumptions that are made for this estimation (the current number of aphid species is treated as the final number, two representative areas are perfectly explored, etc.) are far from the truth, there is a high probability that the real number of Aphidiinae species is several times higher than the obtained result. 

Recently, Engel et al. [6] emphasized that the shortage of brains and hands involved in taxonomy is one of the main reasons for such a large knowledge gap in taxonomy. Further analysis of the data about Aphidiinae taxa gathered since 2010 provides us with exact evidence for their claim. Two out of four genera were described from the Palaearctic (*Sergeyoxys*—Russia, *Astigmapraon*—China) while *Choreopraon* and *Ishtarella* were discovered in New Zealand (Australasia) and Thailand (Indomalayan realm), respectively. The predominance of discoveries in the Palaearctic is much more obvious at a species level, with 70% of species being described from this region. At the same time, none of the species were discovered in Afrotropical and Neotropical regions (Figure 3). In general, Aphidiinae species richness depends on habitat richness and, most importantly, aphid species richness, and is a product of the evolutionary history of the group. As obligatory parasitoids, Aphidiinae follow their hosts in distribution, and consequently the majority of species are found in temperate regions of the Northern Hemisphere [17]. However, such a large difference in species discoveries between Palaearctic and the rest of the world can be largely attributed to the insufficient number of taxonomic experts and their uneven distribution around the world.

The uneven distribution of experts was even more obvious when species authors were analyzed. The 64 Aphidiinae species described between 2010 and 2021 were named by 22 researchers (up to three authors per species), from which five were from the USA (Nearctic), one from India (Indomalayan), and 16 from the Palaearctic (two from China; one each from Japan, Iran, and South Korea; and 11 from Europe). Only nine species were described by authors not from Europe. 

Although the number of species per area/country cannot be used for a biodiversity assessment per se (as explained above), in this particular case it could be very informative as a proxy of uneven distribution of Aphidiinae taxonomy experts (Table 2). Areas or countries are chosen as representatives of a specific continent or region (the largest or most extensively researched). A closer look at those simple data reveals strange patterns in species richness. The only biologically (and biogeographically) logical fact is that the highest number of Aphidiinae species was recorded from Russia (which occupies 30% of the Palaearctic). For example, some of the biggest areas and countries of the world, such as North America, India, and China, all have a similar number of species to two relatively small, landlocked European countries: Czech Republic and Serbia. This discrepancy can be explained only by the lack of Aphidiinae taxonomy experts, because the number of aphid species is much higher, especially in North America [71], where around 1500 aphid species are recorded. Although taxonomists examine specimens from all over the world, most of their work is related to their place of residence, which means those areas are investigated in much more detail.

### 2.3. Aliens in Europe 

Species descriptions and rate of species description are often used as the sole indicator of taxonomic activity, but taxonomy is much more than just naming a species, and various indicators can and should be used to assess the current state of taxonomy and systematics [92]. One such indicator, which is widely underestimated, is the crucial role of taxonomists in the identification of alien species. In European Aphidiinae, only five species were marked as alien (*Aphidius colemani* Viereck, 1912; *Aphidius smithi* Sharma and Subba Rao, 1959; *Lysiphlebus testaceipes* (Cresson, 1880); *Pauesia cedrobii* Starý and Leclant, 1977 and *Pauesia unilachni* (Gahan, 1927)) before 2010, and all were intentionally introduced as biocontrol agents [93,94]. *Lysiphlebus testaceipes* deserves special attention. As a promising biocontrol agent of citrus aphids (*Toxoptera aurantii* (Boyer de Fonscolombe, 1841) and *Aphis spiraecola* Patch, 1914), it was introduced in Europe in 1973 (in the Mediterranean part of France), and very quickly became invasive and widespread over the Mediterranean area [95]. In the last decade, *L. testaceipes* made a breakthrough in the cooler territories of the continental part of Europe [95,96]. On the other hand, the same species was introduced in South Africa (in 1969), where it apparently failed to establish colonies, and also in Kenya, but the fate of the released parasitoids is unknown [80]. Recently, *L. testaceipes* was recorded in high numbers in both Western (Benin) [97] and Eastern Africa (Malawi) [98]. Although its origin is unknown in both countries, it most likely spread naturally from South Africa or Kenya. 

Three more alien species, namely *Lysiphlebus orientalis* Starý and Rakhshani, 2010, *Aphidius ericaphidis* Pike and Starý, 2011 and *Trioxys liui* Chou and Chou, 1993, were detected in Europe in the last decade; all were accidentally introduced [99,100,101]. Interestingly, two species were detected in Europe soon after their descriptions. *Lysiphlebus orientalis* was described in 2010 from China and detected in samples from Serbia collected in 2010 and 2011 [99], while *Aphidius ericaphidis* was described in 2011 from the USA and was detected in samples from Serbia and Scotland collected in 2014 and 2015, respectively [100]. Additional revision of collections indicated that both species were present in Europe long before their formal descriptions (*L. orientalis* in 1995, and *A. ericaphidis* in 1965). *Trioxys liui* was first collected in Spain in 2017 [101]. Furthermore, *Trioxys sunnysidensis* Fulbright and Pike, 2007 was described as a parasitoid of bird cherry-oat aphid (*Rhopalosiphum padi* (Linnaeus 1758)) in Washington State [102]. In 2019, Čkrkić et al. determined that this species is widely distributed in North America and Europe, and also present in New Zealand [88]. Although *T. sunnysidensis* is most likely a cereal aphid parasitoid, it remains undetected because of its rarity. They emphasized that some rare and even new or alien species can be easily overlooked in large-scale ecological studies [88]. 

Considering that Aphidiinae are reducing populations of aphids (which could be treated as pests), some economically oriented (and environmentally unenlightened) policymakers/politicians may conclude that the introduction of alien parasitoids is a good thing. Yet, this could not be further from the truth. Every alien species undoubtedly has negative effects on a new environment. The majority of effects were recently summarized in “Scientists’ warning on invasive alien species” [103]. In this particular case, alien Aphidiinae species can affect the richness and abundance of native species and modify trophic networks, which may lead to changes in ecosystem functioning and the delivery of ecosystem services [103]. For example, alien species can outcompete some native species, resulting in native species’ extinction from occupied areas, or in host shift by native species, which will then trigger a whole cascade of events. At the same time, alien species could have unusual characteristics that reduce their capacity to control aphid populations, such as in the case of the relationship between *L. orientalis* and its host, which is characterized by transgenerational fecundity compensation [104]. Transgenerational fecundity compensation is a phenomenon in which the offspring of parasitized aphids produce more progeny than the offspring of nonparasitized aphids [104]. On the other hand, climate change will inevitably affect local fauna, with local extinctions of native species and/or the appearance of new exotic species. Even relatively small climatic changes can affect aphid–parasitoid communities and result in changes over a relatively short period of time [105].

### 2.4. Shrinking the Gap by Revising the Knowledge

A single description of a species is of inestimable importance, but sometimes alpha-taxonomy [106] can be in discrepancy with beta-taxonomy [106] because of the “superficial description taxonomic impediment” (older descriptions often can be too superficial by today’s standards) [3]. In such cases, a revision of a whole group of organisms (species group, genus, tribe, etc.) is necessary in order to reduce the knowledge gap. 

The majority of Aphidiinae tribes, subtribes, and genera were revised 30–50 years ago and those revisions are mainly outdated. In the last decade, a lot of effort has gone into the improvement of Aphidiinae taxonomy and systematics and the clarification of species status. Revision of the world Monoctonina Mackauer, 1961, is the only revision on the subtribe level, and included all available species [53,57]. Several genera with fewer species were also reviewed: *Monoctonia* Starý, 1962 [42], *Lipolexis* Foerster, 1862 [55], *Areopraon* Mackauer, 1959 [49], *Pseudopraon* Starý, 1975 [49], *Paralipsis* Foerster, 1862 [44,51], *Acanthocaudus* Smith, 1944 [47], *Euaphidius* Mackauer, 1961 [107], and *Remaudierea* Starý, 1973 [107]. Most of those revisions resulted in a higher number of species within the genus, while the genera *Euaphidius* and *Remaudierea* were determined as junior synonyms of *Aphidius* [107]. European species of the genera *Adialytus* Foerster, 1862, [108] and *Lysiphlebus* Foerster, 1862, [50] were also revised, and a new subgeneric classification of European *Ephedrus* Haliday, 1833, species was proposed [52]. *Lysiphlebus* revision [50] can serve as a classical example of the importance of revisions in taxonomy, and consequently in biodiversity research. Prior to revision, there were 15 *Lysiphlebus* species known in Europe. With this study, the number of European *Lysiphlebus* species was reduced to 13; four species were synonymized and two new species were described. Although this quantitative taxonomic information is very important, even more important is the quality of the taxonomic information [92]. Within this study, based on molecular markers and morphology, Tomanović et al. determined that only two descriptions of *Lysiphlebus* species match today’s standards in Aphidiinae taxonomy, and redescribed all other species [50]. In addition, remarks about species biology and distribution were provided. 

There are several published studies dealing with the taxonomic status of Aphidiinae species groups [37,40,41,94,109,110,111,112], among which the most important are those about the taxonomy of biocontrol agents belonging to the *Aphidius colemani* and *A. eadyi* species groups [94,111]. It was determined that the *Aphidius colemani* species group consisted of three species: *A. colemani*, *A. transcaspicus* Telenga, 1958, and the almost forgotten *A. platensis* Brethes, 1913 [111]. At the time when this study was conducted, parasitoids within globally commercially distributed materials were a mixture of all three species [111], and most likely, the situation is the same now. Interestingly, similar results were obtained within the *A. eadyi* species group [94]. Three species of biocontrol agents against *Acyrthosiphon pisum* (Harris) were identified (*A. smithi*, *A. eadyi* Stary, Gonzalez and Hall, 1980, and *A. banksae* Kittel, 2016). *Aphidius banksae*, which was previously known only from Israel and Turkey, was identified as a widely distributed species with a range that covers most of the western Palaearctic [94]. There were also a few studies conducted to test taxonomic status and relationships among three closely related biocontrol agents of cereal aphids, *Aphidius uzbekistanicus* Luzhetzki, 1960, *Aphidius rhopalosiphi* De Stefani Perez, 1902, and *Aphidius avenaphis* (Fitch, 1861) [109,113]. Unlike the previous two studies [94,111], the results of those studies were not straightforward (which is quite common) considering taxonomy. Using molecular and morphological data, the authors determined the incongruence between those two datasets and discovered the possible existence of cryptic species, which could not be morphologically identified and described because of a small number of samples [109,113]. *Aphidius rhopalosiphi* is one of the most studied Aphidiinae species from various aspects of biology, which was also confirmed with this literature survey. High genetic diversity was discovered within *A. rhopalosiphi*, which may imply that it is a species complex, or at least a group of separate evolutionary lineages (which can also differ in biology, ecology, etc.) [109], which calls into question the results of all these studies. Such huge knowledge gaps in the taxonomy of economically very important biocontrol agents raise other questions, especially about the taxonomic status of many other less explored Aphidiinae species.

### 2.5. Keys to Unlock an Easier Scientific Existence 

So-called “bad taxonomy” can have a significant effect on our knowledge of nature [114]. As a bridge that connects and supports many (if not all) biological disciplines, taxonomy needs to be very precise. A single taxonomic error (incorrect identification) may be incorporated in numerous ecological and biological studies, and later even in some environmental programs. With every new step, an error multiplies its impact and could have different negative consequences [114]. In order to reduce the potential for error, taxonomists must provide usable and reliable identification keys. For nontaxonomists, this may seem trivial and easy, but from a taxonomist’s perspective, an identification key is one of the most challenging publications to create. In the last dozen years, there were 40 papers published in which the authors provided identification keys for Aphidiinae genera and species. The studies can be divided according to subject into several groups: (1) identification keys for Aphidiinae from specific (local) areas: from the Middle East and North Africa [86], Iran [33,86 and references therein], Malta [115], Argentina [116], Pakistan [117,118], Australia [78], Costa Rica [119], Serbia [69], and China [120]; (2) identification keys for species within tribes [120], subtribes [53], genera [16,42,44,47,49,50,51,52,55,108,112,121], and species complexes [111]; and (3) identification keys for Aphidiinae species related to specific aphids [30], plants [32,36,122,123,124,125,126,127], and habitats [128]. 

Although a significant number of Aphidiinae identification keys have been published in the last 12 years, there is still a significant shortage, primarily because of species coverage in existing taxonomic keys. Currently, we lack identification keys even for areas that are relatively well explored. There are several very usable keys that cover European species of a specific subtribe or genus, e.g., [50,51,53,55]. We are still lacking keys for some of the most difficult and problematic genera (*Aphidius*, *Praon*, *Pauesia*, and *Trioxys*). The most comprehensive identification key is one about Aphidiinae in Serbia (covers 121 species) [69]; unfortunately, it is published only in the Serbian language. At the same time, users of taxonomic end-products (such as species descriptions and identification keys) largely avoid using them, or at least do not cite such work. Bortolus [114] found that 62.5% of papers published in top-ranked ecological journals are missing any information about the literature used for the identification of the organisms in the study.

## 3. Looking in and through the Mirror—Current Situation and Future Prospects 

Taxonomy is one of the most undervalued biological disciplines [6]. It is safe to say that the current state of Aphidiinae taxonomy is better than in most parasitoid groups, but it is far from satisfactory. The taxonomy of Aphidiinae is facing the same problems as taxonomy in general. The knowledge gap is still very large, as illustrated in the previous sections. Considering the number of experts, subfamily Aphidiinae is in a more favorable situation than other subfamilies and families of parasitoids. Ten taxonomy experts (Željko Tomanović (Belgrade, Serbia), Elena Davidian (St. Petersburg, Russia), José Manuel Michelena Saval (València, Spain), Nickolas Kavallieratos (Athens, Greece), Ehsan Rakhshani (Zabol, Iran), Keith Pike (Washington State, USA), Jelisaveta Čkrkić (Belgrade, Serbia), Korana Kocić (Belgrade, Serbia), Maryna Kaliuzhna (Kyev, Ukraine), and myself) for a relatively small group of parasitoids looks great, but their distribution is highly inconvenient: 8 out of 10 researchers are from Europe, and half of them are from the same research team (University of Belgrade Faculty of Biology). Fortunately, there are a few young Aphidiinae taxonomists who will replace retired pioneers Petr Starý, Manfred Mackauer, and Ulf Gärdenfors. Thus, although there are numerous cracks in the mirror’s surface, the reflection still looks good and promising, and we can say that the current state of Aphidiinae taxonomy is on its way to becoming almost satisfactory. 

Now, it is time to look through the mirror. Future prospects for Aphidiine taxonomy are the same as for taxonomy in general. There are ever-increasing demands for scientific names [129], while the rate of naming species is constant or just slightly increasing. In 2010–2021, Aphidiinae species were described at a rate of 5.33 species per year on average. For almost two decades, we have been witnessing different attempts to accelerate the taxonomic process and numerous debates for and against some of the proposals. The two most “revolutionary” proposals are, in fact, technological approaches in which species descriptions should be replaced with DNA barcodes [130,131], while type specimens should be replaced with photographs of species taken in the field [131,132]. From its beginnings, taxonomy has been integrative, but species descriptions are based on a set of characters (with emphasize character state), and, in most cases, illustrated with line drawings and/or (later) photographs. In a broader context, those “revolutionary” approaches are just simplifications of taxonomy in the way of using just one character (DNA barcodes) instead of many [6], and keeping the illustration, but not the voucher specimens. Researchers who advocate those ideas give many different reasons why it is “better” than traditional taxonomy. For example, Minteer et al. stated that voucher specimens should be replaced with a “series of good photographs, which can even be used to describe a species, complemented by molecular data and a description of a species’ mating call for birds, amphibians, or insects” in order to avoid the extinction that can be caused by collecting [132]. There are at least two questionable aspects of this statement. Firstly, if you are studying small insects (less than 5 mm), and the majority of insects are small, it is highly improbable that one will be able to take a series of high-resolution photos in the field (sometimes you cannot even take one, if you manage to see the insect in the first place). In addition, in most cases, characters for species’ identification are very small body parts such as the genitals, tarsal claws, tibial spurs, etc. The equipment for taking good photographs of those characters is too robust and heavy for fieldwork, and the insect needs to be still for photographing. After that, a DNA sample should be taken. Everyone with experience with small insects (e.g., parasitoids) knows that most small insects do not survive such disturbance, and then we will get an unwanted voucher specimen. Secondly, if collecting one or few specimens can threaten species survival, that species will most probably go extinct in the blink of an eye anyway, and then we have lost out on an opportunity to gather knowledge. 

We often hear that DNA barcoding is becoming cheaper and cheaper, and thus affordable to all. Those who write those statements are simply unable to see the insects for the pipette tips. If you look beyond your comfort zone, you will realize that at least half of the scientific community can access basic scientific literature only thanks to a modern Robin Hood figure named Alexandra Elbakyan (creator of the website Sci-Hub, which provides free access to research papers without regard for copyright). For example, back in 2007, Godfray wrote that DNA sequencing was becoming cheaper and more affordable [131], but in order to read his article (and find out what was so cheap) you needed to pay $32, which represents a significant proportion of the average monthly salary in some parts of the world.

The idea behind the Barcode of Life Initiative is excellent, but the authors of the initial paper [130] wrote, “When fully developed, a COI identification system will provide a reliable, cost-effective and accessible solution to the current problem of species identification.” Currently, we are still far from a fully developed system, and yet there are some studies that exclusively use DNA barcodes for species diagnosis in Braconidae [133]. There is already a wide debate considering the justification of such minimalist revisions [3,6,134,135]. Meier at al. stated that such studies will become the next “superficial description impediment”, which is probably the strongest argument against this practice [3]. Indeed, taxonomic diagnoses should be clear, and involve a minimal number of statements which will allow us to distinguish a given specimen from other taxa [135]. Although Sharky et al. revised 11 Braconidae subfamilies, Aphidiinae were not analyzed [133].

DNA barcoding is a widely accepted method in Aphidiinae taxonomy, but only as a part of an integrative taxonomic approach that (in most cases) employs DNA barcodes along with morphological and ecological data. In the last two years, 10 Aphidiinae species have been described thanks to cooperation between taxonomists and the Barcode of Life Initiative [53,55,57], and even more species were described using the DNA Barcode as part of an integrative approach. All three alien species recorded in Europe since 2010, together with *T. sunnysidensis*, were also identified by integrative taxonomy. In this small scientific community, the opinion prevails that using all available types of data is the only way to shrink the knowledge gap in taxonomy.

In the last few years, different researchers started using DNA barcodes for the molecular identification of Aphidiinae [79,98,136]. The identification of Aphidiinae based solely on molecular data (barcodes) is not reliable, because species and genera boundaries, based on barcoding sequences, vary significantly. For example, within the genus *Ephedrus*, there are two groups of species that differ by as much as 20% (the genetic distance between *E. persicae* and *E. plagiator* clade is 20.7%) [52]. On the other hand, within the genus *Aphidius*, genetic distances between species are much lower, in some cases less than 1.5% [137], and within *Lysiphlebus* there are species with an even lower genetic distance. There are several documented cases where DNA barcodes fail to discriminate species that are morphologically and ecologically different, such as *Aphidius ervi* Haliday, 1834 and *Aphidius microlophii* Pennacchio and Tremblay, 1987, as well as *A. uzbekistanicus* and *A. avenaphis*. Considering the aforementioned, it is almost impossible to use only molecular markers in Aphidiinae species identification. Barcodes can be used as guidance for identification, but it is still obligatory to perform morphological identification to confirm species identity. This brings us to the importance of voucher specimens. In ecological studies, the use of the fastest DNA extraction protocols is still common practice, even if they are destructive and leave no voucher specimens. The aforementioned studies about the *Aphidius colemani* group in Eastern Africa [98,136] and the native Aphidiinae of New Zealand [68] used destructive DNA extraction protocols and lost potentially very valuable information, while those from New Zealand probably also lost several as yet undescribed species. There are a number of nondestructive protocols, e.g., [138,139,140] that could and should be used for voucher specimens.

If we manage to keep the current core of Aphidiinae taxonomists and engage some more from different parts of the world, there is a relatively bright future for Aphidiinae taxonomy. Another important task would be closer cooperation with other researchers, especially ecologists. This collaboration could bring about many interesting and more precise results that will give us a better understanding of our world.

## 4. Conclusions

The bridge from the beginning of the story looks a bit different after a decade of research. Some holes have been filled, and some obstacles have been removed. Now, we can see better. We can see some new holes, obstacles, and gaps that need to be fixed.

Even with the tremendous work that has been completed so far, the knowledge gap in Aphidiinae taxonomy is still significant, and all aspects of the taxonomic impediment are obvious.

Although there is no meaningful research in biology without reliable taxonomy [60], its importance in the modern world is far from fully acknowledged [141]. Recently, the Swiss Re Institute report concluded that 55% of global gross domestic product (GDP) is dependent on biodiversity and ecosystem services [142], but we are still lacking large research grants (and even small ones) for taxonomy, which is considered only as a cost [141]. Until institutions, governments, and the world realize the importance of this kind of research, taxonomists need to be cleverer, and utilize data from all available resources such as museum collections, as well as cybertaxonomy [143], and molecular data.

Anyone who intends to take this interesting walk over the bridge called taxonomy should be aware that in Aphidiinae taxonomy, barcodes are not enough, and must be used only as a part of integrative taxonomy. A similar situation is seen for most other taxa, and using just one type of information can make our knowledge gaps even larger.

It seems appropriate to finish with a citation of a song by Jonathan Coulton (https://theportalwiki.com/wiki/Still_Alive_(song) (accessed on 10 June 2021)) adapted for taxonomy:

“We’ve experiments to run

There is research to be done

On the species who are

Still alive.”

## Figures and Tables

**Figure 1 insects-13-00170-f001:**
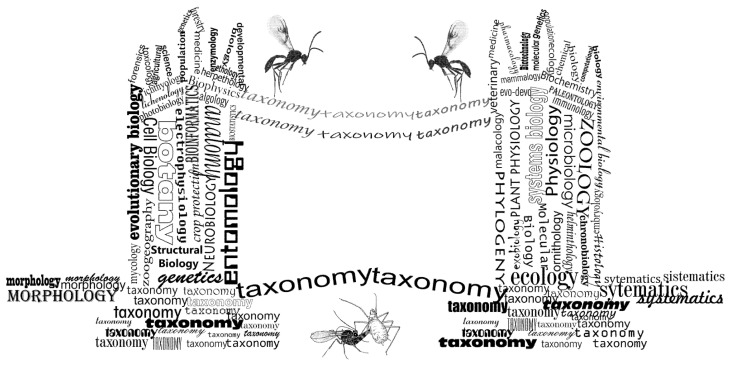
Position of taxonomy in biology today (inspired by *words are bridge* by mariateresa.dapolito).

**Figure 2 insects-13-00170-f002:**
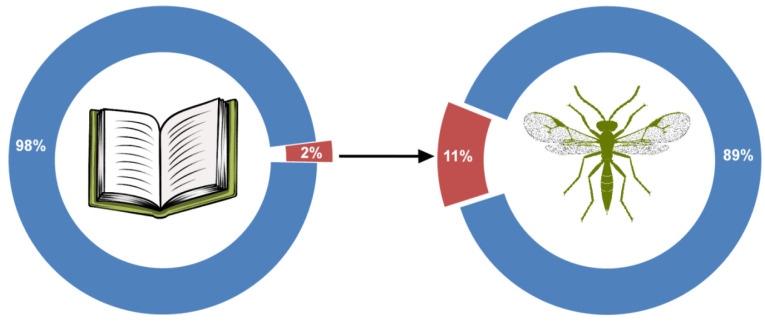
Share of taxonomic papers (red) in Aphidiinae studies conducted in the period 2010–2021 (**left**) and share of new Aphidiinae species described in the same period (red) compared to the total number of Aphidiinae species (**right**).

**Figure 3 insects-13-00170-f003:**
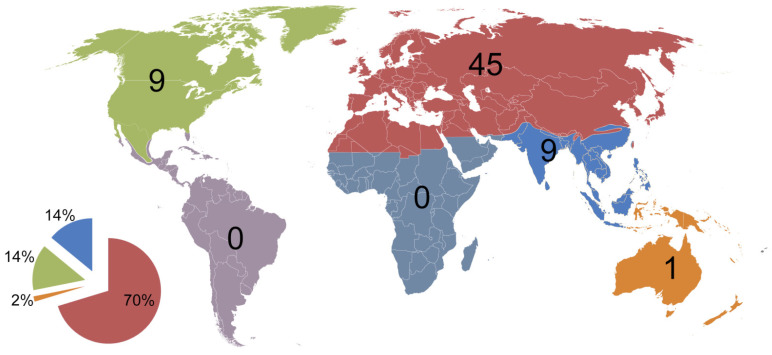
Number and percentage (pie chart) of Aphidiinae species described in different biogeographic realms in 2010–2021.

**Table 1 insects-13-00170-t001:** Results of literature surveys for Aphidiinae studies using Google Scholar (GS), Scopus, and Web of Science (WoS).

Search System	Total Number of Results	Number of Relevant Results	Number of Unique Results Included in Analysis
**GS**	3570	1753	1752
**Scopus**	874	803	125
**WoS**	1082	654	25
**Total**	5526	3209	**1902**

**Table 2 insects-13-00170-t002:** Number of Aphidiinae species recorded in some countries/areas, with references in square brackets.

Country/Area	Number of Species	References	Area (km^2^)
**America north of Mexico** (Nearctic)	~130	[47,57,58,72,73]	19,782,990
**Mexico** (North and Central America)	33	[74]	1,972,550
**Costa Rica** (Central America)	10	[75]	51,100
**Chile** (Neotropics)	23	[76]	756,096.3
**Brazil** (Neotropics)	19	[77]	8,515,767
**Australia** (Australasia)	23	[78]	7,692,024
**New Zealand** (Australasia)	15	[79]	268,021
**Subsaharan Africa** (Afrotropics)	22	[80,81,82,83,84]	23,290,000
**Madagascar**	7	[84]	592,800
**Russia** (Palaearctic)	198	[54,61,85]	17,098,246
**Middle East and North Africa**	108	[86]	11,695,164
**China** (Asia)	130	[16]	9,596,961
**India** (South Asia)	127	[29,34]	3,287,263
**Japan** (Far East)	≈80	[42,55,64,87]	377,975
**Kyrgyzstan** (Central Asia)	35	[60]	199,951
**Norway** (Northern Europe)	26	[58,88,89]	385,207
**Czech R****epublic** (Central Europe)	≈135	[21]	78,871
**Germany** (Central and Western Europe)	109	[88,90]	357,022
**Great Britain** **and Ireland** (British Isles)	96	[88,91]	293,752
**Serbia** (Southern Europe)	121	[69]	88,361

## Data Availability

The dataset (list of studies) generated during the current study is available from the author on request.

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
