# Peer review of "Sizing the Knowledge Gap in Taxonomy: The Last Dozen Years of Aphidiinae Research"

_insects, 2022, doi:10.3390/insects13020170_

Round 1

Reviewer 1 Report

The subject manuscript is a literature review combined with commentary and criticism regarding a variety of topics relevant to taxonomy and systematics. In my view, the literature review is substantially flawed, both in terms of methodology and presentation (see specifics below, particularly my comments regarding lines 94-138). The commentary and criticism are not substantial enough to merit publication on their own.

The manuscript will require significant English language revision, primarily for grammar and punctuation. Though meaning is generally clear, there are some cases where sentences become factually incorrect or ambiguous due to grammar issues.

Lines 32-34: Who is ignorant, and what, exactly, are they ignorant about? The author just finished stating that the biodiversity crisis is starting to get the attention that it deserves from the public.

Lines 35-36 and Figure 1: Describing taxonomy as a “bridge” between morphology and systematics is not accurate in my view. First, it ignores the ever-expanding role of molecular data in both taxonomy and systematics. Second, it fails to capture the relationship between taxonomy and systematics: a systematic taxonomy is one in which the taxonomy is consistent with common ancestry as assessed by systematic studies, which may be morphology or DNA-based. Figure 1 offers no clarification on the issue, and only further muddles the relationships among various biological disciplines. It’s not clear what the two columns are supposed to represent (certainly not “morphology” and “systematics”). Many subdisciplines are present in both columns (including simply “biology”). Why is “Botany” on the left, and “Malacology” on the right? In what sense are aphidiine braconids a bridge between all these disparate disciplines? Is the author suggesting that veterinary medicine is somehow linked to biophysics through aphidiine braconids?

Line 70: While it is fine to laud the accomplishments of Mackauer and Starý, it is very unseemly to call them the “real pioneers” in contrast to the list of earlier entomologists. It would be more appropriate to say something like “Contemporary scientists, such as Mackauer and Starý, have continued to build on the work of these early pioneers…”

Lines 76-78: At the conclusion of the introduction, it is not at all clear how a review of taxonomic studies of a single wasp family can serve as “proxy” for the overall taxonomic knowledge gap. Perhaps “case study” would be a better term?

Lines 79-89: In order to demonstrate the disparate counts of aphidiine species, three studies are cited: Boivin et al.(400 species), Zikic et al. (505 species), and Mackaur and Finlayson (700 species). Of these, only Zikic et al. cite the Taxapad World Ichneumonidae database (developed by Dicky Yu) as the source for their species count. It seems strange, then, to cite “uncritical use” of Yu’s database as a source of error when the cited number from Yu (505 species) is so close to the estimated number (500 species, based vaguely on “available data”) provided by the author. These “available data” need to be cited, especially given that they form the foundation for the genus and species counts used to calculate proportional increases due to newly described taxa (see lines 107-109) and to build Figure 2. Finally, the discrepancies in estimated numbers of species, and especially numbers of genera, could partially be the result of the push and pull of the taxonomic splitters vs. lumpers, a possibility not mentioned by the author.

Line 94: There are many problems with using Google Scholar to perform a systematic search of scientific literature. Google Scholar results are based on data submitted to Google by publishers. Publishers that communicate heavily with Google will be over-represented in the results, and those that don’t may be omitted entirely despite being perfectly reputable (scholarly) resources. Also, Google’s algorithms, and the content that they search, are changing constantly and without notice. A Google search is not reproducible, and therefore not appropriate as the primary methodology for a peer-reviewed scientific paper. In my view, this issue alone renders the subject manuscript unsuitable for publication. Literature reviews of this type should be based upon curated databases of scientific sources (e.g. Web of Science, PubMed, Agricola, etc.).

Lines 105-111: How do these numbers by themselves indicate a knowledge gap in aphidiine taxonomy? We need an estimate of the total number of species worldwide (described + undescribed) to demonstrate any sort of knowledge gap.

Lines 134-138, including Table 1: I do not doubt that certain regions have received a disproportionate amount of attention from taxonomists (this is true for most insect groups). However, more information is needed about how the table was constructed. Is the table a comprehensive list of every country/region where Aphidiinae are known to occur? If yes, then it is a stark demonstration that certain countries/regions have received disproportionate taxonomic attention; it would be remarkable indeed if 10 species were known from Costa Rica, and zero were known from all the remaining Central American countries. This, however, does not appear to be the case. I know, for example, that some Aphidiinae species are known from Mexico, but Mexico is not included in the table. The author should describe what criteria were used to include a given county/region in the table.

Line 153: 2.2 Foreigners in Europe (New alien species in Europe): Why the double title for this section? If this is an attempt at cleverness, it instead sounds xenophobic. I suggest creating a title that links the taxonomic knowledge gap to invasive species ecology. The three paragraphs in this section summarize the status of some invasive Aphidiinae in Europe, but these cases should be more clearly tied back to the concept of a taxonomic knowledge gap. The author seems to be suggesting that a lack of taxonomic expertise might allow an invasive species to go unidentified (and therefore undetected as an invasive). However, that doesn’t seem to be the case for the presented examples.

2.3 Revising the unrevised: This section should be retitled to establish a link between the taxonomic knowledge gap and revisionary work.

2.4. Keys to unlock an easier scientific existence (life). this section title is intriguing, but ultimately non-sensical in light of the paragraphs that follow it. Suggest deleting the parenthetical “(life)”.

Section 3. Looking in and through the mirror – Current state and future prospect. Maybe change the title to “Looking in and beyond the mirror…”? This section is a somewhat polemical description of the author’s views on DNA barcoding, DNA-based descriptions, DNA and photograph-based specimen vouchering, and the Open Access movement. Many good points are made, but none are particularly novel. The section would also benefit from improved focus and organization. The examples from Aphidiinae do not add unique or interesting dimensions to the arguments.

Reviewer 2 Report

Review of MS 1561473: Sizing the knowledge gap in taxonomy: last dozen years of Aphidiinae research

The MS addresses an interesting topic concerning the taxonomic study gap of an important family of parasitoid wasps such as the subfamily Aphidiinae (Hymenoptera: Braconidae). In addition to the specific topic dealt with, the author highlights the overall importance of taxonomy, especially in this delicate moment for biodiversity, for studies that concern all aspects of plant and especially animal biology.

Line 15-17. “Aphidiinae….” Move the sentence to the next paragraph;

Line 27. Correct “parasiotoids”;

Line 40. Geography…??? Etc.;

Figure 1. Position of taxonomy in biology today. La figura è evocative può l’autore scirvere tra parantesi l’autore della stessa;

Line 53-56. Does the author have a bibliographic reference or a study concerning the orders in which the unknown species concern the hymenoptera and / or diptera? For example, thrips and many beetles are very small, so from where you can deduce what is written;

Line 59. Are all Aphidiinae parasitoids? Or are there also hyperparasitoids and more?

Line 79. The author therefore confirms that in the last twelve years only 3 papers describing the Aphidiinae have been published;

Line 82. I think it is superfluous to indicate this value (43%);

Line 96-109. The discrepancy that the author points out for this group is evident as the applicative aspects related to the biological control of different species are evident for the group so using the relationship between what is published in the total and the papers of the systematic seems to me not well representative of the reality of the different insects groups;

Line 121-124. Also include information on the distribution of Aphidiinae hosts;

Line 144-146. This sentence is not very clear and we do not understand the role and who is the biggest country;

In paragraph 2.1 it would be desirable to report in which of the countries reported there is the greatest number of known species of aphids known. This to find a further link between the hosts and parasitoids often associated to this group of hosts;

Line 153. In this paragraph it is desirable to include some information on the relationships between several species of parasitoids and the climatic factors linked to global warming or adverse events that can cause complex relationships of several species (see https://doi.org/10.3390/insects11120841);

Line 204. Reference for alpha and beta-taxonomy;

Line 216. Correct “nuber”;

Line 384-385. Insert references on non-destructive DNA extraction methods on arthropod specimens;

Line 302-308. Reducing the sentence seems very long and gives no additional information;

Line 404.  GDP, also write this abbreviation in exstenso;

Line 409. Do you have any reference for cybertaxonomy;

Reviewer 3 Report

Manuscript Insects-1561473 titled “Sizing the knowledge gap in taxonomy: last dozen years of Aphidiinae research” is a well written review that flow well from start to end emphasizing the importance of taxonomy and support of taxonomists of this important biocontrol agents.  I recommend consideration for publication in Insects MDPI in the present form. I made some annotations for author’s revision in the attached PDF. Some notes are:

  • A list of taxonomists with their affiliation and recent contact information can be useful as a supplementary table which may help identification of species of Aphidiinae for researchers and students in Africa and Latin America and probably encourage new students for training and courses.
  • Author explains the problems facing taxonomy researchers of Aphidiinae and the lack of Aphidiinae taxonomy experts in many regions of the world. What the authors suggest helping mitigate this problem?
  • What the author suggests for the valuable non-English research work on Aphidiinae that are written in Spanish, Serbian, Polish, and Russian languages. Can these be accurately translated by Google translator?  

Round 2

Reviewer 1 Report

The author has done a phenomenal job addressing my issues with the literature search methodology, and as such this manuscript now represents a strong contribution and review of literature search methodologies. 

Regarding my comments about the subtitle "Foreigners in Europe (New alien species in Europe)," please understand that I am not saying that you had xenophobic intentions, or that you are personally xenophobic. The problem arises from the subtlety of language, and the fact that two nearly synonymous phrases are linked into a single subtitle. The term "foreigner" typically refers to people, whereas "alien species" typically refers to non-human invasive species. Therefore, the subtitle implicitly links "foreign people" to "invasive species" (generally not considered a good thing). Given that I do not think this was your intention, I thought it prudent to point out how this subtitle might be interpreted.

Author Response

Dear Sir/Madame,

Thank you very much for clarification of previous comments. Also, thank you once more for valuable comments which improved my manuscript.